# CONIC10K: A Challenging Math Problem Understanding and Reasoning Dataset

**Haoyi Wu**[◇1,2], **Wenyang Hui**[◇1,2], **Yezeng Chen**[1,3,6], **Weiqi Wu**[†7], **Kewei Tu**[♣1,2], **Yi Zhou**[♣3,4,5]

[1]School of Information Science and Technology, ShanghaiTech University
[2]Shanghai Engineering Research Center of Intelligent Vision and Imaging
[3]School of Information Science and Technology, University of Science and Technology of China
[4]National Engineering Laboratory for Brain-inspired Intelligence Technology and Application
[5]Key Laboratory of Brain, Cognition and Education Sciences, Ministry of Education
[6]Shanghai Innovation Center for Processor Technologies
[7]Department of Computer Science and Engineering, Shanghai Jiao Tong University
{wuhy1, huiwy, chenyz, tukw}@shanghaitech.edu.cn;
wuwq1022@sjtu.edu.cn; yi_zhou@ustc.edu.cn

## Abstract

Mathematical understanding and reasoning are crucial tasks for assessing the capabilities of artificial intelligence (AI). However, existing benchmarks either require just a few steps of reasoning, or only contain a small amount of data in one specific topic, making it hard to analyse AI's behaviour with reference to different problems within a specific topic in detail. In this work, we propose **CONIC10K**, a challenging math problem dataset on conic sections in Chinese senior high school education. Our dataset contains various problems with different reasoning depths, while only the knowledge from conic sections is required. Since the dataset only involves a narrow range of knowledge, it is easy to separately analyse the knowledge a model possesses and the reasoning ability it has. For each problem, we provide a high-quality formal representation, the reasoning steps, and the final solution. Experiments show that existing large language models, including GPT-4, exhibit weak performance on complex reasoning. We hope that our findings could inspire more advanced techniques for precise natural language understanding and reasoning. Our dataset and codes are available at https://github.com/whyNLP/Conic10K.

## 1 Introduction

Mathematical understanding and reasoning ability is an important component of human intelligence. Such an ability is the foundation of data analysis, financial applications and scientific research. Though there have been lots of studies

---

◇ Equal Contribution.
♣ Corresponding Authors.
† Work completed while the author was at ShanghaiTech University.

(Lample and Charton, 2020; Wei et al., 2022b), mathematical reasoning are far from being solved by existing methods (Lu et al., 2022), even with symbolic reasoners (Hopkins et al., 2019) and large language models (LLMs) (Lightman et al., 2023). To evaluate and analyse the mathematical ability, various datasets and benchmarks have been proposed in recent years (Zhao et al., 2020; Hendrycks et al., 2021; Mishra et al., 2022b,a). However, these datasets or benchmarks often suffer from the following problems: (1) The problems can be solved with only a few reasoning steps, so language models may rely on shallow heuristics to achieve high performance (Patel et al., 2021); (2) The dataset covers a wide range of topics and hence there is only a small amount of data for each topic, which makes it hard to distinguish whether the model fails because of a lack of background information, or due to weak reasoning ability.

To address the above issues, we propose **CONIC10K**, an open-ended math problem dataset on conic sections in Chinese senior high school education. This dataset contains 10,861 carefully annotated problems, each one has a formal representation, the corresponding text spans, the answer, and natural language rationales. Figure 1 shows an example problem in our dataset. To evaluate the mathematical understanding and reasoning ability, we perform two different tasks on existing LLMs: semantic parsing and mathematical question answering (mathQA). Semantic parsing assesses a language model's ability to understand mathematics. The model is required to translate the math problem in natural language into its formal meaning representations. MathQA jointly evaluates the language model's ability of mathematical understanding and reasoning. The model needs to gen-

erate the answers to questions. Since the topic of **CONIC10K** is restricted to conic sections, the knowledge required to solve different problems is the same, while the only difference is the difficulty in reasoning. Therefore, if the model is able to solve simple problems but not hard ones, we are assured that the failure lies in the lack of ability in mathematical reasoning.

Our experiments show that current models obtain good performance in semantic parsing. However, in mathQA, these models are far from being satisfactory. When performing zero-shot chain of thought (CoT) (Wei et al., 2022b) prompting, the best model **GPT-4** (OpenAI, 2023) can only achieve 15.5% accuracy using human evaluation. When finetuning is further applied, the best model **ChatGLM-6b** (Du et al., 2022) still obtains a poor accuracy of 22.5% under human evaluation. When we translate the problems into English and apply zero-shot CoT to reason in English, the accuracy of **GPT-4** is 26.0%, which is still far below the performance of human experts at 57.5% with a 3-minute time limit for each problem. This shows that the poor performance is not due to the language being used but to a deficiency in reasoning ability. Therefore, we believe the mathematical reasoning ability of language models is still limited despite their huge success in natural language understanding.

We conclude our contributions as follows: 1) We propose **CONIC10K**, a challenging math problem dataset on conic sections in Chinese senior high school education, with high-quality annotations of formal representations; 2) We perform experiments to inspect the mathematical understanding and reasoning ability of LLMs separately; 3) We give detailed analysis on the model behaviour and conduct comprehensive case studies. We hope that our work could help the community to better analyse LLMs in mathematical understanding and reasoning and inspire more advanced techniques to enhance the mathematical reasoning ability of LLMs.

## 2 Related Work

There has been a wide range of datasets on math problems in the literature. MATHQA (Amini et al., 2019) and GSM8K (Cobbe et al., 2021) are math word problem datasets. They focus on open-domain understanding, where the objective is to extract a single equation based on the information about quantities in the problem, rather than mathematical reasoning. Similarly, Math23K

(Wang et al., 2017) and Ape210K (Zhao et al., 2020) are popular datasets about Chinese math word problems with open-domain scenarios and simple reasoning steps. Geometry3K (Lu et al., 2021) is a geometry problem-solving dataset that provides formal representations, but the dataset size is small and the problems do not require complex reasoning. AQuA (Ling et al., 2017), NumGLUE (Mishra et al., 2022b) and Lila (Mishra et al., 2022a) are large-scale datasets of various math problems. They have been used as benchmarks in solving math word problems and mathematical reasoning tasks, but we find that these datasets require only a few reasoning steps. MATH (Hendrycks et al., 2021) is the one with the longest reasoning steps among these datasets. It has been used as a standard benchmark in recent work of LLMs (Lewkowycz et al., 2022; Lightman et al., 2023). However, while it covers a wide range of problems, it contains limited data in each specific topic, making it hard to analyse the model behavior in detail with reference to one topic. It also does not provide any formal representations. Our proposed **CONIC10K** contains problems of long reasoning steps using closed-domain knowledge and has high-quality annotations with formal representations. A detailed comparison between the aforementioned datasets and **CONIC10K** is shown in Table 1.

## 3 Dataset

### 3.1 Formal Representation

We design a formal representation that avoids ambiguity and is close to natural language. Specifically, our representation is built upon Assertional Logic (Zhou, 2017). Assertional Logic (AL) is a powerful knowledge representation that is more expressive than first-order logic while easier to read and write for humans. In this work, we use a variant of AL with three components: declarations, facts and queries. Declarations define individuals with their types (e.g. `G:Ellipse`). Facts are assertions that describe the conditions in the problem (e.g. `Focus(G)={F1, F2}`). Queries are the terms that represent the goal of the problem (e.g. `Range(Eccentricity(G))`). See more details in Appendix A.

### 3.2 Dataset Format

An example is presented in Figure 1. For each question, we give 1) the question text in natural language with math formulas in LaTeX, 2) the ra-

| Question: | Formal Representation: | Span: |
| --- | --- | --- |

**Question:**

点$P(x,y)$是椭圆$\frac{x^2}{a^2}+\frac{y^2}{b^2}=1\,(a>b>0)$上的任意一点，$F_1,F_2$是椭圆的两个焦点，且$\angle F_1PF_2\leq 90°$，则该椭圆的离心率的取值范围是？

*(Let $P(x,y)$ be an arbitrary point on the ellipse $\frac{x^2}{a^2}+\frac{y^2}{b^2}=1\,(a>b>0)$. $F_1$ and $F_2$ are the two foci of the ellipse, and $\angle F_1PF_2\leq 90°$. What is the range of values for the eccentricity of the ellipse?)*

**Rationale:**

由题意可知，当点$P$位于$(0,b)$或$(0,-b)$处时，$\angle F_1PF_2=90°$最大，此时 $\cos\angle F_1PF_2=\frac{a^2+a^2-4c^2}{2a^2}=\frac{a^2-2c^2}{a^2}\geq 0, a\geq\sqrt{2}c$。因为$e=c/a$，所以$e\leq\frac{\sqrt{2}}{2}$。因为$e$是椭圆离心率，$0<e<1$，所以$0<e\leq\frac{\sqrt{2}}{2}$。

*(When the point $P$ is located at $(0,b)$ or $(0,-b)$, the angle $\angle F_1PF_2\leq 90°$ is at its maximum. In this case, $\cos\angle F_1PF_2=\frac{a^2+a^2-4c^2}{2a^2}=\frac{a^2-2c^2}{a^2}\geq 0, a\geq\sqrt{2}c$. Since $e=\frac{c}{a}$, we have $e\leq\frac{\sqrt{2}}{2}$. As $e$ represents the eccentricity of the ellipse, and it lies within the range $0<e<1$, we can conclude that $0<e\leq\frac{\sqrt{2}}{2}$.)*

**Answer:**

$(0,\frac{\sqrt{2}}{2}]$

| Formal Representation | Span |
| --- | --- |
| `P: Point` | 点$P(x,y)$ |
| `PointOnCurve(P, G)=True` | 点$P(x,y)$是椭圆…上的任意一点 |
| `Coordinate(P)=(x1, y1)` | $P(x,y)$ |
| `x1,y1: Number` | $P(x,y)$ |
| `G: Ellipse` | 椭圆$\frac{x^2}{a^2}+\frac{y^2}{b^2}=1\,(a>b>0)$ |
| `Expression(G)=(y^2/b^2+x^2/a^2=1)` | 椭圆$\frac{x^2}{a^2}+\frac{y^2}{b^2}=1\,(a>b>0)$ |
| `a, b: Number` | $\frac{x^2}{a^2}+\frac{y^2}{b^2}=1\,(a>b>0)$ |
| `a > b` | $\frac{x^2}{a^2}+\frac{y^2}{b^2}=1\,(a>b>0)$ |
| `b > 0` | $\frac{x^2}{a^2}+\frac{y^2}{b^2}=1\,(a>b>0)$ |
| `F1, F2: Point` | $F_1,F_2$ |
| `Focus(G)={F1, F2}` | $F_1,F_2$是椭圆两个焦点 |
| `AngleOf(F1,P,F2)<=Unit(90,degree)` | $\angle F_1PF_2\leq 90°$ |
| `Range(Eccentricity(G))=?` | 该椭圆的离心率的取值范围是？ |

Figure 1: Example problem from the **CONIC10K** dataset.

| Dataset | Size | Language | Formal Rep. | Rationale | Reasoning Steps |
| --- | --- | --- | --- | --- | --- |
| **AQuA** (Ling et al., 2017) | 100,000 | English | ✗ | Natural Language | 2.15 |
| **Math23K** (Wang et al., 2017) | 23,162 | Chinese | ✗ | Equation | 1.59 |
| **MATHQA** (Amini et al., 2019) | 37,297 | English | ✗ | Program | 2.99 |
| **Ape210K** (Zhao et al., 2020) | 210,488 | Chinese | ✗ | Equation | 2.02 |
| **GSM8K** (Cobbe et al., 2021) | 8,792 | English | ✗ | Natural Language | 2.25 |
| **Geometry3K** (Lu et al., 2021) | 3,002 | English | ✓ | ✗ | 2.57 |
| **MATH** (Hendrycks et al., 2021) | 12,500 | English | ✗ | Natural Language | 4.65 |
| **NumGLUE** (Mishra et al., 2022b) | 101,835 | English | ✗ | ✗ | 1.67 |
| **Lila** (Mishra et al., 2022a) | 134,000 | English | ✗ | Program | 1.70 |
| **Conic10K (Ours)** | 10,861 | Chinese | ✓ | Natural Language | 4.23 |

Table 1: Comparison of our **CONIC10K** dataset with existing datasets. **CONIC10K** is the largest dataset that has formal representation annotated. It is also the dataset that has the second longest average number of reasoning steps in all languages and has the longest average number of reasoning steps in Chinese.

tionale in natural language, 3) the answer to the question, 4) the formal representation and 5) the text span corresponding to each sentence in the formal representation.

## 3.3 Dataset Construction

### 3.3.1 Data Collection

To construct the dataset, we first collect approximately 20,000 open-ended problems about conic sections from two websites that focus on Chinese high school education in image format. Each problem image contains the problem text, rationale, and answer. Then, we use mathpix[1] to convert these images into text. Since our dataset is focused on conic sections, we filter out problems that involve knowledge from other topics such as sequences and solid geometry. After that, we remove duplicated

problem using fuzzy matching. After the above process is finished, the size of the dataset is reduced from around 20,000 to approximately 14,000.

### 3.3.2 Annotation

To ensure the correctness of the data and avoid ambiguities, we apply strict quality control during the annotation process[2]. The complete process is as follows:

**Initiation** We first build a small dataset with hundreds of samples, write the annotation guidelines and design a rule-based AI assistant for annotation. The rule-based AI assistant is able to recognize LaTeX math expressions and complete simple formal representations, which greatly accelerates the annotation process and reduces annotation errors.

[1] https://mathpix.com/

[2] See Appendix A.3 for more details.

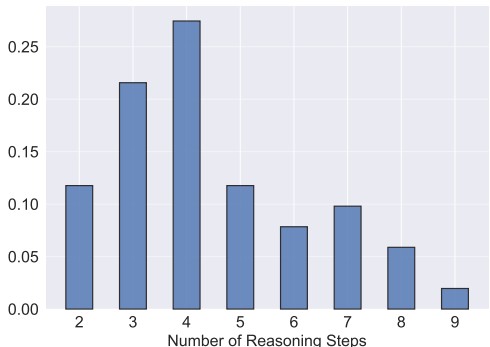

Figure 2: Distribution of reasoning steps in 50 sampled problems from **CONIC10K**. All numbers are rounded to their nearest integers.

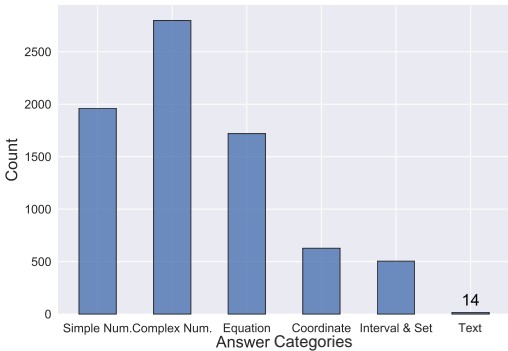

Figure 3: Distribution of the 7,758 training examples on answer categories.

| | |
|---|---|
| Num. problems | 10,861 |
| Num. operators | 94 |
| Num. concepts | 20 |
| Avg. LaTeX expressions in a problem | 5.76 |
| Avg. tokens in a problem | 83.43 |
| Avg. sentences in a problem | 3.41 |
| Avg. sentences in formal rep. of a problem | 10.55 |
| Avg. operators in formal rep. of a problem | 15.70 |
| Avg. individuals in a formal rep. of a problem | 4.51 |

Table 2: Statistics about **CONIC10K**. Problems are tokenized using bert-base-chinese tokenizer[4] in Avg. tokens in a problem.

**Verification**  We select the annotators from a group of candidates by their performance on the small dataset. These annotators are provided with annotation guidelines along with hundreds of samples. Annotators with the best performance will take part in the rest of the annotation process.

**Annotation**  We ask the annotators to further filter out problems about other topics, write the formal representation, select the corresponding text spans and fix the incorrectly recognized problem texts and answers. Each problem is annotated by two annotators, and then validated by another validator with an automated tool for comparison. We also randomly check 3% of the annotations. This process takes 4 months in total.

**Finalization**  After the annotation is finished, we train a language model[3] through 5-fold cross-validation, manually check the inconsistency between model predictions and the annotated formal representations, and fix the errors in annotations. This helps us correct another 2% of the data. Then we randomly split the dataset into train, validation, and test sets with the ratio 7.5:1:2. The train set size is 7,758, the validation set size is 1,035, and the test set size is 2,068. We proceed to the evaluation of LLMs with this split.

### 3.4 Dataset Statistics

Table 2 presents the basic statistics about **CONIC10K**. The problems in our dataset tend to be long and complex. Besides these metrics, we also estimate the number of reasoning steps by the minimum number of rules required to get enough information to obtain an answer. Since the process of applying rules is subjective, we ask two graduate students to individually annotate the rules used to solve the problems. We uniformly sampled 30 problems from each of the datasets listed in Table 1 and ask the two students to annotate the reasoning steps. Results show that **CONIC10K** is the dataset with the second largest number of reasoning steps. The distribution of reasoning steps in **CONIC10K** is depicted in Figure 2. We show additional dataset statistics in Appendix B.

To facilitate model analysis, we divide the answers into 6 categories as described in Table 3. Figure 3 shows the distribution on these categories.

## 4 Experiments

This section describes our experiments to evaluate the mathematical understanding and reasoning abilities of various models.

### 4.1 Tasks

Based on data provided by **CONIC10K**, we introduce two tasks: **semantic parsing** and **mathQA**.

---

[3]We finetune the **OPUS-mt-zh-en** (Tiedemann and Thottingal, 2020). It is a machine translation model that translates Chinese into English.

| Category | Examples | Description |
|---|---|---|
| Simple Number | $2, -1$ | Numerical values composed of a single number |
| Complex Number | $1/3, \sqrt{5} - 1$ | Numerical values composed of multiple numbers |
| Equation | $x^2 + y^2/4 = 1$ | Equations |
| Coordinate | $(0, 1), (-\sqrt{2}, 0)$ | Coordinates of points |
| Interval & Set | $[-1, 1], \{0, 1\}$ | Intervals and sets |
| Text | 'ellipse' | Texts |

Table 3: Answer categories with examples and description.

Semantic parsing requires a model to translate math problems in natural language into formal representations, while mathQA needs a model to give correct solutions to math problems. The semantic parsing task aims solely at assessing the model's ability to understand mathematics, and the mathQA task jointly evaluates the model's ability of mathematical understanding and reasoning.

## 4.2 Models

We evaluated the performance of several popular pretrained models on the above two tasks. The models used for evaluation are as listed in Table 4.

## 4.3 Evaluation Details

Due to limited computation resources, we conducted full finetuning on models with size of less than 4B. For models around 7B, we performed parameter efficient finetuning using LoRA (Hu et al., 2022) and 8-bit quantization (Dettmers et al., 2022). We also apply zero-shot CoT inference without finetuning for models with sizes between 7B and 13B. The models evaluated in zero-shot CoT setting all have undergone instruction tuning or RLHF in their respective pretraining process. When finetuning, we use instruction tuning (Wei et al., 2022a) to train the models. The instructions are architecture-specific and task-specific, as depicted in Table 5.

When finetuning language models, we use the following hyperparameter settings. We use AdamW as the optimizer. The learning rate is selected from $\{8e-5, 2e-5\}$, with a linear learning rate decay. For models using LoRA, we set target modules to $q, k, v$ for **Falcon-7b** and to $q, v$ for other models. The LoRA rank is set to 16 for models with size around 7B. To ensure a similar number of trainable parameters, we set the LoRA rank to 24 for **Bloomz-3b** and 32 for **Bloomz-1b7**. We use greedy decoding in all generations.

In zero-shot CoT inference for mathQA, we use the same prompt as GAOKAO-Benchmark (Zhang et al., 2023) to instruct the models to give an answer together with a rationale. In MathQA, we also experiment with in-context learning (Min et al., 2022), which adds in-context demonstrations of the task in the prompt, and self-consistency (Wang et al., 2023), which conducts majority voting on the sampled results on **GPT-3.5-turbo**. In semantic parsing, however, the formal representation is unknown to the above models. Since it requires more than 3,000 tokens to explain the syntax and semantics of each component in the formal language, which is out of the context length limit of most models listed above, we do not evaluate the performance of zero-shot CoT in semantic parsing.

In addition to the methods mentioned above, we also evaluate the following two methods in mathQA as a reference: **(1) Guessing '2'**: Predicting the most frequent answer in the train set, which is '2'. **(2) Human Experts**: We randomly select 20 problems from the test set and ask two graduate students to answer. Each problem has a 3-minute time limit. We report the average accuracy of these two students.

## 4.4 Metrics

### 4.4.1 Semantic Parsing

For semantic parsing, we evaluate the model predictions by micro-F1, macro-F1 and accuracy. The accuracy is the proportion of the problems that have a one-to-one match between all sentences in the prediction and the ground truth. Micro-F1 (mi-F1) and macro-F1 (ma-F1) are defined as follows:

$$\text{mi-F1} = 2 \cdot \frac{pr}{p + r}, \tag{1}$$

$$\text{ma-F1} = \frac{\sum_{i=1}^{n} \text{F1}_i}{n} \tag{2}$$

where $n$ is the total number of problems, $p = \frac{\text{\# of all matched sentences}}{\text{\# of all predicted sentences}}$ is the overall precision, $r = \frac{\text{\# of all matched sentences}}{\text{\# of all gold sentences}}$ is the overall recall, $\text{F1}_i$ is the F1 score of problem $i$.

To compute the metric, we need to find the number of matched sentences between the prediction and ground truth. Since the formal representation is insensitive to individual naming, we enumerate all possible individual name mappings between prediction and ground truth and select the mapping

[5]https://chat.openai.com/, we use **GPT-3.5-turbo-0314** version.

[6]we use **GPT-4-0314** version.

| Model | Sizes | Architecture | Base Model | Chinese-Oriented | IT & RLHF |
|---|---|---|---|---|---|
| **mT5** (Xue et al., 2021) | 300M-13B | Encoder-decoder | - | ✗ | ✗ |
| **mT0** (Muennighoff et al., 2022) | 300M-13B | Encoder-decoder | **mT5** | ✗ | ✓ |
| **LLaMA** (Touvron et al., 2023) | 7B-65B | Decoder-only | - | ✗ | ✗ |
| **Vicuna** (Chiang et al., 2023) | 7B, 13B | Decoder-only | **LLaMA** | ✗ | ✓ |
| **Ziya** (Yang et al., 2022) | 13B | Decoder-only | **LLaMA** | ✓ | ✓ |
| **Bloom** (Scao et al., 2022) | 560M-176B | Decoder-only | - | ✗ | ✗ |
| **Bloomz** (Muennighoff et al., 2022) | 560M-176B | Decoder-only | **Bloom** | ✗ | ✓ |
| **ChatGLM** (Du et al., 2022) | 6B | Prefix Decoder | - | ✓ | ✓ |
| **Falcon** (Penedo et al., 2023) | 7B, 40B | Decoder-only | - | ✗ | ✗ |
| **Falcon-inst** (Penedo et al., 2023) | 7B, 40B | Decoder-only | **Falcon** | ✗ | ✓ |
| **GPT-3.5-turbo**[5] | ? | Decoder-only | - | ✗ | ✓ |
| **GPT-4**[6] (OpenAI, 2023) | ? | Decoder-only | - | ✗ | ✓ |

Table 4: Models used in our experiments. Chinese oriented refers to whether methods, such as increasing the portion of Chinese data and designing a tokenizer for Chinese, are used to improve performance in Chinese tasks. **IT** stands for instruction tuning and **RLHF** stands for reinforcement learning with human feedback.

| Architecture | Task | Instruction |
|---|---|---|
| Encoder-decoder | SP | Please translate the following problem into expressions: "*problem*" |
| Encoder-decoder | MQA | Please give an answer to the following problem: "*problem*" |
| Decoder-only | SP | The translation into expressions of "*problem*" is |
| Decoder-only | MQA | The answer to "*problem*" is |

Table 5: Instructions used in finetuning. *problem* is replaced by the problem text when training.

that achieves the maximum number of matched sentences. We optimize the evaluation script by only considering individuals with the same type so that the evaluation time on the validation set and test set is acceptable.

### 4.4.2 MathQA

In mathQA, since it is nontrivial to automatically determine whether two answers are the same (e.g., $1/\sqrt{2}$ vs. $\sqrt{2}/2$, $x - y = 0$ vs. $x = y$, and $3x + 4y = 5$ vs. $\frac{3}{5}x + \frac{4}{5}y - 1 = 0$), we rely on human evaluation to determine the correctness of model answers.

## 5 Results and Discussions

In this section, we introduce and explain the results of the experiments. The main results of semantic parsing and mathQA are shown in Table 6 and Table 7 respectively.

### 5.1 Semantic Parsing

Language models show good ability of understanding on math problems after proper training.

| Model | Trainable Param. | mi-F1 | ma-F1 | Acc. | # Syntax Err. |
|---|---|---|---|---|---|
| *Finetuning PLM* | | | | | |
| **mT5-base** | 580M | 93.1 | 93.7 | 66.3 | 19 |
| **mT0-base** | 580M | 95.8 | 96.2 | 77.2 | 10 |
| **mT5-large** | 1.2B | 95.8 | 96.2 | 77.6 | 12 |
| **mT0-large** | 1.2B | 96.7 | 96.9 | 80.7 | 6 |
| **mT5-xl** | 3.7B | 96.9 | 97.2 | 82.6 | 9 |
| **mT0-xl** | 3.7B | **97.4** | **97.5** | **84.6** | 8 |
| *Finetuning LLM using LoRA* | | | | | |
| **Bloomz-1b7** | 7M | 90.0 | 90.7 | 62.7 | 13 |
| **Bloomz-3b** | 7M | 91.5 | 92.2 | 67.6 | 6 |
| **Bloomz-7b1** | 8M | 94.3 | 94.7 | 71.3 | 4 |
| **Falcon-7b** | 12M | 89.5 | 89.6 | 58.0 | 10 |
| **LLaMA-7b** | 8M | 94.0 | 94.8 | 71.1 | 5 |
| **ChatGLM-6b** | 8M | 95.1 | 95.8 | 74.7 | 7 |
| **Vicuna-7b** | 8M | 96.2 | 96.6 | 76.9 | **3** |

Table 6: Results on semantic parsing in CONIC10K. The fully finetuned **mT0-xl** achieve the highest accuracy, while the LoRA finetuned **Vicuna-7b** achieves the lowest syntax error rate.

The best model **mT5-xl** can successfully translate 84.6% of the problems into formal representations. For the problems it fails to accurately translate, the predictions only differ from the ground truth in minor details. The F1 score and accuracy from **Bloomz** family and **Falcon-7b** are much lower than other models. The performance of finetuned instruction tuned models is consistently better than that of finetuned base models.

**Models pretrained on code show strong ability in learning syntax.** Models except for the **mT5** family have been pretrained on code. The syntax error rates of these models are on average lower than that of the **mT5** family, even though their F1

score and accuracy may be lower than the **mT5** family. Since the formal representation resembles programming languages in syntax, pretraining on code may be able to help model to learn the syntax of formal representations more easily.

**Increasing model size effectively improves model's performance in semantic parsing.** From the results of the model families **mT5, mT0** and **Bloomz**, we find that increasing the model size from the smallest to largest in our experiment can significantly improve the accuracy by at least 7.4%.

## 5.2 MathQA

Language models generally show poor performance on mathQA in CONIC10K. Under the zero-shot CoT setting, most models achieve an accuracy close to 0. Even after finetuning, the accuracy of the best model is still significantly lower than that of human experts by 35.0%.

**Simple problems under finetuning setting may not be simple under zero-shot CoT setting.** Most models finetuned on CONIC10K have the best performance in **Simple Numbers** among the answer categories. However, when it comes to zero-shot CoT setting, **GPT-4** and **GPT-3.5-turbo** obtain best accuracy in **Coordinate**. One possible reason is that after sufficient training on CONIC10K, the model can develop a shallow understanding of the task (Patel et al., 2021), including the frequent answers of a specific kind of questions. Since **Simple Numbers** are simpler in form and have fewer potential answers compared to **Coordinates**, being familiar with the answer distribution can effectively increase the probability to hit the correct answer. However, in zero-shot CoT setting, the model is unaware of these distributions, so it has no advantage in difficult problems that have simple answers.

**The accuracy is close to 0 in zero-shot CoT.** Under the zero-shot CoT setting, **Bloomz-7b1** and **Falcon-7b-inst** show extremely poor performances with 0 accuracy in all problems. These models tend to generate repetitive patterns, and in most cases fail to give an answer. Other models except for **GPT-4** generate text that looks like a valid rationale, but the majority of reasoning steps are incorrect. They often produce hallucinations in premises and rules, and derive wrong results. In Table 9, even with in-context demonstrations or majority voting, the performances are still low. We showcase some failing cases in Table 10.

**The scaling law is less clear compared to semantic parsing.** Though we observe that increasing the model size continuously and effectively improves model performance in semantic parsing, such a phenomenon disappears in mathQA tasks. In **mT5** and **mT0** series, large models do not necessarily outperform small models. Similar observations have been made in MATH (Hendrycks et al., 2021) where the authors find that accuracy on math problems increases only modestly with model size.

**Chinese-oriented language models have better performance in mathQA in CONIC10K.** In the zero-shot CoT setting, the two Chinese-oriented models, **Ziya-13b** and **ChatGLM-6b**, achieve the best performance below **GPT-3.5-turbo**. In the finetuning using LoRA setting, **ChatGLM-6b** achieve an accuracy of 22.5% and outperform other models by a large margin.

**Translating problems into English does not make the performance of GPT-4 on par with human experts in mathQA.** We translate the problems into English and evaluate **GPT-4** in zero-shot CoT setting to determine whether the poor performance is due to language or long reasoning steps. The results in Table 8 show the performance is significantly improved from 15.5% to 26.0% by translating the problems into English. However, this accuracy is still low compared to 57.5% from human experts. Therefore, the primary challenge of mathQA in CONIC10K still lies in how to do mathematical reasoning correctly.

## 5.3 Case Study

We inspect and analyse both success and failure cases in the experiment, which leads us to some interesting findings.

**LLMs have limited ability in understanding long LATEX expressions.** 9.7% of the incorrect predictions from **mT0-xl** are due to errors in translating simple but long LATEX expressions. Common failures include missing terms, flipped signs and incorrect copies. For example, the LATEX expression in the problem is `x^2+y^2+2\sqrt{2}x-4\sqrt{2}y+10-r^2=0`, but the translated sentence becomes `-4*sqrt(2)*y +2*sqrt(2)*x+x^2+y^2+2=-r^2`. In this example, we observe both a flipped sign and an incorrect constant. We do not observe similar errors in relatively short LATEX expressions.

| Model | Trainable | Accuracy of Answer Category | | | | | | |
|---|---|---|---|---|---|---|---|---|
| | Param. | Simple Num. | Complex Num. | Expression | Coordinate | Interval & Set | Text | All |
| *Finetuning PLM* | | | | | | | | |
| **mT5-base** | 580M | 5.1 | 11.5 | 8.4 | 5.0 | 1.6 | 0.0 | 7.2 |
| **mT0-base** | 580M | 22.8 | 13.6 | 7.3 | 2.5 | 4.9 | 0.0 | 13.0 |
| **mT5-large** | 1.3B | 21.0 | 14.8 | 8.1 | 3.7 | 4.9 | 0.0 | 13.0 |
| **mT0-large** | 1.3B | 16.7 | 17.0 | 12.5 | 3.7 | 6.6 | 0.0 | 13.8 |
| **mT5-xl** | 3.7B | 19.9 | 17.6 | 11.0 | 5.0 | 6.6 | 0.0 | 14.8 |
| **mT0-xl** | 3.7B | 18.1 | 13.6 | 10.3 | 2.5 | 6.6 | 0.0 | 12.5 |
| *Finetuning LLM using LoRA* | | | | | | | | |
| **Bloomz-1b7** | 7M | 23.2 | 0.0 | 0.0 | 0.0 | 0.0 | 0.0 | 6.3 |
| **Bloomz-3b** | 7M | 26.1 | 7.6 | 8.1 | 3.7 | 1.6 | 0.0 | 12.0 |
| **Falcon-7b** | 16M | 31.5 | 4.8 | 8.4 | 15.0 | 8.2 | 0.0 | 14.0 |
| **Bloomz-7b1** | 8M | 27.9 | 11.8 | 12.5 | 6.2 | 3.3 | 0.0 | 15.4 |
| **LLaMA-7b** | 8M | 34.1 | 9.1 | 9.9 | 8.7 | 4.9 | 0.0 | 15.8 |
| **Vicuna-7b** | 8M | 37.7 | 9.4 | 12.8 | 10.0 | 8.2 | 0.0 | 17.9 |
| **ChatGLM-6b** | 8M | **39.3** | **23.1** | 13.1 | 10.6 | **6.5** | 0.0 | **22.5** |
| *Zero-shot CoT* | | | | | | | | |
| **Bloomz-7b1** | - | 0.0 | 0.0 | 0.0 | 0.0 | 0.0 | 0.0 | 0.0 |
| **Falcon-7b-inst** | - | 0.0 | 0.0 | 0.0 | 0.0 | 0.0 | 0.0 | 0.0 |
| **Vicuna-7b** | - | 1.5 | 0.0 | 0.0 | 0.0 | 0.0 | 0.0 | 0.4 |
| **Vicuna-13b** | - | 3.1 | 0.4 | 0.0 | 0.0 | 0.0 | 0.0 | 0.9 |
| **Ziya-13b** | - | 2.8 | 0.9 | 0.7 | 0.0 | 0.0 | 0.0 | 1.1 |
| **ChatGLM-6b** | - | 4.0 | 0.7 | 0.2 | 1.3 | 0.0 | **14.3** | 1.5 |
| **GPT-3.5-turbo** | - | 8.5 | 4.6 | 4.0 | 12.3 | 0.6 | **14.3** | 6.2 |
| **GPT-4** | - | 17.8 | 11.8 | **20.4** | **21.4** | 5.3 | 0.0 | 15.5 |
| *References* | | | | | | | | |
| **Guessing '2'** | - | 18.1 | 0.0 | 0.0 | 0.0 | 0.0 | 0.0 | 4.5 |
| **Human Expert** | - | 62.5 | 56.3 | 50.0 | 50.0 | 66.7 | - | 57.5 |

Table 7: Results on mathQA in **Conic10K**. **ChatGLM-6B** achieves the best overall accuracy after finetuning using LoRA among all the models. In fully finetuning setting, **mT0-xl** shows strongest performance. In the zero-shot CoT setting, **GPT-4** has the highest overall accuracy. However, the performances of the above models are significantly lower than human expert's performance. **GPT-4** is evaluated on 200 randomly sampled problems. **Human Expert** is evaluated on 50 randomly sampled problems. The **Text** accuracy of **Human Expert** is empty because the sampled problems do not contain answers of category **Text**.

| Language | Overall Accuracy |
|---|---|
| **GPT-3.5-turbo + CoT** | 6.2 |
| **GPT-3.5-turbo + CoT + ICL** | 5.9 |
| **GPT-3.5-turbo + CoT + SC** | 6.8 |

Table 8: Results on mathQA in **Conic10K** using **GPT-3.5-turbo** with in-context-learning (ICL) or self-consistency (SC)

| Language | Overall Accuracy |
|---|---|
| Chinese | 15.5 |
| English | 26.0 |

Table 9: Results on Chinese problems and problems translated to English in mathQA in **Conic10K** using **GPT-4** with zero-shot CoT. Both are evaluated on the same 200 sampled problems.

**Models can hardly find shortcuts in reasoning in mathQA.** We observe that models usually employ naive approaches to solve problems and fail to find shortcut solutions, which leads to more complicated computation and longer reasoning steps. The additional reasoning steps and computation make the models more likely to make mistakes during reasoning. Some examples of naive solutions from **GPT-4** and the corresponding shortcut solutions are listed in Table 11 and 12.

**GPT-4 and GPT-3.5-turbo probably lack knowledge about certain concepts.** When asked problems about focal distance, **GPT-4** and **GPT-3.5-turbo** keep giving incorrect answers and often give a value that is half of the ground truth. Based on these observations, we suspect that these two models lack knowledge about focal distance. We ask **GPT-4** and **GPT-3.5-turbo** to explain what focal distance is in both Chinese and English, and they

keep defining it as the distance between the center of an ellipse or hyperbola and one of its foci instead of the correct definition, the distance between the two foci. A probable reason is that 'focal distance' is not a commonly used term within the English corpus, making the models unlikely to obtain correct knowledge about it.

## 6 Conclusion

We present CONIC10K, a math problem understanding and reasoning benchmark. It provides problems that require complex reasoning, while only involving knowledge about conic sections in Chinese senior high school education. We test popular LLMs on both semantic parsing and math question answering, inspecting model performance and behaviours. Results show that existing LLMs, including **GPT-4**, have poor performance in mathematical reasoning, while most models could achieve good performance in mathematical understanding (but not perfect yet). We analyse the model predictions in detail and find LLMs tend to hallucinate in reasoning, often fail to find the shortcuts solution, and may lack the knowledge to solve problems. We hope our dataset, CONIC10K, can help to discover the weaknesses of LLMs in mathematical understanding and reasoning and inspire more advanced techniques to enhance the mathematical reasoning ability of LLMs.

## Limitations

CONIC10K is a dataset with high-quality formal representation annotations, but there are still some limitations:

- We design the formal representation to be accurate, unambiguous and close to natural language, but such representation is not commonly used and does not fit any existing symbolic reasoners. The conclusion may not apply to other formal representations such as propositional logic and first-order logic, or rationales like executable programs.

- In conic sections, the commonly used mathematical reasoning strategies could be limited. For example, our problems may require solving simultaneous equations systems, but not likely mathematical inductions. Therefore, our dataset cannot evaluate some reasoning strategies such as mathematical induction.

## Ethics Statement

CONIC10K is a dataset that requires massive data sources and heavy annotation. We claim that our work is free of ethical risks from the following perspectives:

**Data Source** The problems in CONIC10K are collected from two websites that do not limit the usage of data for education and research purpose. We strictly follow the term of use and manually check all the data to avoid inappropriate information in the annotation stage.

**Annotation** We hire a group of 14 annotators for formal representation annotation and sign a contract to prescribe the rights from both sides. We clearly state the purpose of our study and the future data use. These annotators are well-paid for their work. The authors take responsibility to maintain the annotation website, provide necessary documents, answer questions from the annotators and clean up the data.

## Acknowledgements

This work was supported by the National Natural Science Foundation of China (61976139, 62250057), and by Shanghai Frontiers Science Center of Human-centered Artificial Intelligence and MoE Key Lab of Intelligent Perception and Human-Machine Collaboration (ShanghaiTech University).

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

## A  Formal Representation

### A.1  The Assertional Logic

Assertional Logic (AL) (Zhou, 2017) is a formal representation where all kinds of knowledge are formalized by equality assertions. It builds upon the equality properties and the set theory. AL representations are human-friendly and it has been proved that the expressiveness of AL is stronger than first-order logic (or $k$th-order logic for any $k \geq 1$).

Here, we briefly introduce the syntax of AL. Given a specific domain, the syntactic structure of AL is composed of three components: individuals, concepts and operators. Individuals represent objects in the domain, concepts represent groups of objects and operators represent relationships and connections among individuals and concepts. Operators are similar to functions and predicates in first-order logic (FOL), but they could accept high-order constructs (concept, concept of concepts), which leads to the strong expressiveness of AL.

An assertion is of the form $a = b$, where $a, b$ are two terms (individuals, either atomic or compound). The knowledge base of AL is just a set of assertions.

## A.2 Our Representation

We apply AL as our formal representation because of its strong readability. Our principle is that the formal representation should 1) avoid ambiguity. The formal representation should resolve the ambiguity in natural language and with the information inside the annotations, it should be possible to work out the solution by hand; 2) close to natural language. It should be able to represent the problem without rephrasing it; 3) simple and clear. Designing a representation with thousands of operators is definitely expressive and powerful, but it sacrifices the strength of logic and fails to extract common knowledge underneath.

Therefore, we apply only 94 operators and 20 concepts (see Table 2) to represent all the problems in the dataset. To better accommodate the natural language, we also designed 3 pseudo operators: OneOf, WhenMin, WhenMax. These operators do not fit the semantics of AL, but greatly simplify the representation and are closer to natural language. Also, it is trivial to convert these operators to terms in AL.

There also has been evidence showing that rephrasing significantly impacts learning (Kwiatkowski et al., 2013). To avoid rephrasing, we write detailed documents for the annotators, ask them to raise questions when they are not confident and frequently check the data during annotation.

We design our representation in three components: declarations, facts and queries.

**Declarations**  The declarations define individuals with their types. It has the format of var: type, where var is an individual and type is a concept. These sentences are a special representation of the

assertion Is(var, type) = True. For simplicity, we allow defining multiple individuals in one sentence, with commas separating different individuals.

**Facts**  The facts are assertions that describe the conditions in the problem. For clarity, we allow the use of syntactic sugar, which includes $<, \leq, >, \geq, +, -, \times, \div, a^b$. That is, a sentence could be an inequality such as a > b, which indicates an assertion (a > b) = True.

**Queries**  The queries are the terms that represent the target of the problem. They ought to be an assertion with the left-hand-side(LHS) the query term and the right-hand-side(RHS) an unknown individual in AL, but we use the simplest format during the annotation.

## A.3 Annotation Quality Control

Our previous study shows that the annotation of formal language is extremely hard for humans. It is difficult for an experienced annotator to reach an accuracy above 50%. As a result, we employ multiple measures to control the dataset quality, including:

1. We provide a rule-based AI assistant to complete most of the annotations with high precision.

2. We only hire annotators with the highest performance on the small dataset we built in advance.

3. During the annotation, we ask the annotators to raise questions whenever they are not confident about how to annotate. We provide detailed documents and dedicated help to ensure the correctness of the annotation.

4. In addition to formal representations, we ask the annotators to annotate the text spans. We find it helps to increase the annotation accuracy.

5. Each problem will be annotated by two annotators individually, then passed to another validator. We design a web UI which could automatically compare two annotations and extract the difference. The validator will determine which one is correct, or a third annotation is required.

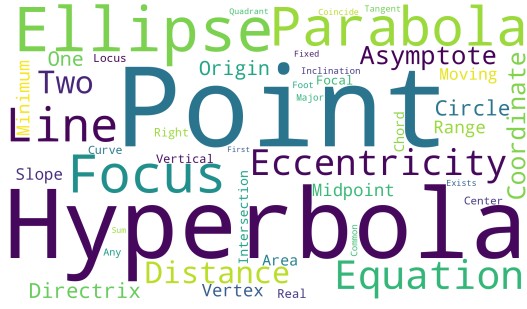

Figure 4: Word cloud of the keywords in **Conic10K**

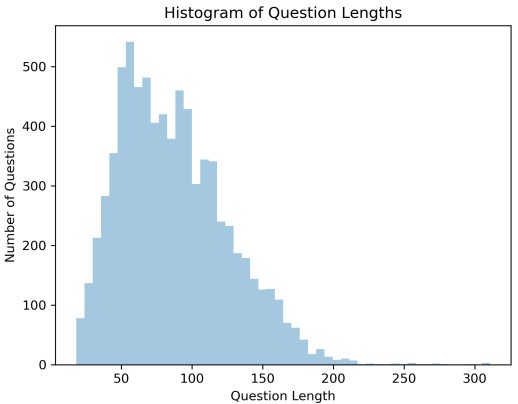

Figure 5: Distribution of Question Length

6. Every time the annotators finish 1000 annotations, we randomly sample 10 problems for additional checks. After all the annotations were finished, we randomly sample 200 problems for additional checks. In the additional check, we independently annotated the sampled problems, and then compare them with the existing annotation. We ask the annotators to do a thorough check if the accuracy is below 80%[7].

7. We provide competitive payments ($> 150k$ CNY in total, $\approx 20k$ USD) to the annotators. We allow adequate time for the annotation process.

8. After the annotation is finished, we finetune a zh-en translation model for further validation. We split the whole dataset into five random splits of the same size. Then, we pick four of them to finetune the model and collect predictions for the last split. We manually check all the problems whose prediction does not match the annotation. We repeat this process five times and obtain the final dataset.

## B  Additional Dataset Statistics

We show the frequency of keywords in Figure 4, and the distribution of question length in Figure 5. In question length, we count all latex commands such as \frac, \leftarrow as one token.

## C  Additional Results

We showcase some interesting outputs from models. Models including **Bloomz-7b** and **ChatGLM-6b** often generate redundant text and hallucinate when reasoning. We provide some samples in Table 10. **GPT-4** shows a strong ability to perform reasoning

in natural language, but its solution often fails to find the shortcuts designed in the problems. Such a solution often requires complicated calculations. Due to its limited ability in arithmetic calculation, the final answer from **GPT-4** is likely to be incorrect. We provide some samples in Table 11 and 12.

---

[7]In the final check, about 99% of the 200 sampled formal representation annotations (without considering text spans) pass the check.

| Problem | Rationale & Solution |
|---|---|
| Given that $M$ is a point on the parabola $x^2 = 4y$, $F$ is its focus, and point $A(1, 5)$, what is the minimum value of $\lvert MF \rvert + \lvert MA \rvert$? | **Redundant solution by Bloomz-7b:** $\lvert MF \rvert + \lvert MA \rvert = \lvert MA \rvert + \lvert MF \rvert + \lvert MA \rvert + \lvert MF \rvert + \lvert MA \rvert + \lvert MF \rvert + \lvert MA \rvert + \lvert MF \rvert + \lvert MA \rvert + \lvert MF \rvert + \lvert MA \rvert + \lvert MF \rvert + \lvert MA \rvert + \lvert MF \rvert + \lvert MA \rvert + \lvert MF \rvert + \lvert MA \rvert + \lvert MF \rvert + \lvert MA \rvert + \lvert MF \rvert + \lvert MA \rvert + \lvert MF \rvert + \lvert MA \rvert + \lvert MF \rvert + \lvert MA \rvert \ldots$ |
| If the eccentricity of an ellipse is $\frac{1}{2}$ and one of its foci coincides with the focus of the parabola $y^2 = 8x$, then what is the standard equation of the ellipse? | **Hallucination in reasoning by ChatGLM-6b:** According to the problem, we know that the eccentricity of the ellipse is $\frac{1}{2}$, and one of its foci coincides with the focus of the parabola $y^2 = 8x$. First, let's find the coordinates of the focus of the parabola $y^2 = 8x$. According to the definition of a parabola, the x-coordinate of the focus is $\frac{p}{2}$, where $p$ is the focal length. Since the standard equation of the parabola is $y^2 = 4px$, we can determine that $p = 2$. Therefore, the x-coordinate of the focus is $\frac{2}{2} = 1 \ldots$ |

Table 10: Translated failing cases. The red text is the reasoning step where hallucination takes place.

| Problem | Rationale & Solution |
|---|---|
| Given a point P on the parabola with equation $y^2 = 4x$, let's denote the distance from point P to the y-axis as $d$. For a given point $A(4, 5)$, what is the minimum value of $\lvert PA \rvert + d$? | **Naive solution by GPT-4:** First, we can write the equation of the parabola as $y^2 = 4x$. Let's assume the coordinates of point P as $(x_0, y_0)$, so we have $y_0^2 = 4x_0$. The distance from point P to the y-axis is denoted as d, which is equal to $d = x_0$. We need to find the minimum value of $\lvert PA \rvert + d$, which is equivalent to finding the minimum value of $\lvert PA \rvert + x_0$. Using the distance formula between two points, we have $\lvert PA \rvert = \sqrt{(x_0 - 4)^2 + (y_0 - 5)^2}$. Substituting $y_0^2 = 4x_0$, we get $\lvert PA \rvert = \sqrt{(x_0 - 4)^2 + (4x_0 - 5)^2}$. Now we need to find the minimum value of $\sqrt{(x_0 - 4)^2 + (4x_0 - 5)^2} + x_0$. To simplify the calculation, we can first find the minimum value of $(x_0 - 4)^2 + (4x_0 - 5)^2 + x_0^2$. Expanding $(x_0 - 4)^2 + (4x_0 - 5)^2 + x_0^2$, we get $17x_0^2 - 40x_0 + 41$. We can find the extreme points by taking the derivative. Differentiating $17x_0^2 - 40x_0 + 41$, we get $34x_0 - 40$. Setting the derivative equal to 0, we find $x_0 = \frac{20}{17}$. Substituting $x_0 = \frac{20}{17}$ into $17x_0^2 - 40x_0 + 41$, we get the minimum value as $\frac{144}{17}$. Therefore, the minimum value of $\lvert PA \rvert + d$ is $\boxed{\sqrt{\dfrac{144}{17}}}$ |
| | **Reference shortcut solution:** Since the distance between a point on a parabola and its focus is the same as the distance between the point and directrix of the parabola, $\lvert PA \rvert + d$ can be rewritten into $\lvert PA \rvert + d_1 - d_2 = \lvert PA \rvert + \lvert FP \rvert - d_2$, where $d_1$ is the distance between $P$ and the directrix, $d_2$ the distance between the directrix and the y-axis, $F$ is the focus. By the definition of parabola, $d_2 = 1$, $F = (1, 0)$. Since $\lvert PA \rvert + \lvert FP \rvert - d_2 \geq \lvert AP \rvert - d_2 = \sqrt{(4 - 1)^2 + (5 - 0)^2} - 1 = \sqrt{34} - 1$. Therefore, the minimum value of $\lvert PA \rvert + d$ is $\boxed{\sqrt{34} - 1}$. |

Table 11: Translated example of solutions from **GPT-4** and shortcut solutions. The red text is the reasoning step where the solution goes wrong.

| Problem | Rationale & Solution |
|---|---|
| The equation of the line that passes through the point $M(3, -1)$ and bisects the chord of the hyperbola $\frac{x^2}{4} - y^2 = 1$ is? | **Naive solution by GPT-4:** First, we need to find two symmetric points on the hyperbola. Let these points be $A(x_1, y_1)$ and $B(x_2, y_2)$, which are symmetric with respect to the point $M(3, -1)$. According to the properties of symmetric points, we have: |

$$x_1 + x_2 = 2 \cdot 3 = 6$$

$$y_1 + y_2 = 2 \cdot (-1) = -2$$

Since points $A$ and $B$ lie on the hyperbola, they satisfy the equation of the hyperbola:

$$\frac{x_1^2}{4} - y_1^2 = 1$$

$$\frac{x_2^2}{4} - y_2^2 = 1$$

Substituting $y_1 = -2 - y_2$ into the first equation, we get:

$$\frac{x_1^2}{4} - (-2 - y_2)^2 = 1$$

Substituting $x_1 = 6 - x_2$ into the above equation, we get:

$$\frac{(6 - x_2)^2}{4} - (-2 - y_2)^2 = 1$$

This is a system of linear equations in terms of $x_2$ and $y_2$. By solving for $x_2$ and $y_2$, we can obtain the coordinates of points $A$ and $B$.

Solving, we find $x_2 = \frac{6 \pm \sqrt{2}}{2}$ and $y_2 = \mp\sqrt{2} - 1$. Therefore, $A\left(\frac{6+\sqrt{2}}{2}, -\sqrt{2} - 1\right)$ and $B\left(\frac{6-\sqrt{2}}{2}, \sqrt{2} - 1\right)$.

Next, we need to find the equation of the line passing through points $A$ and $B$. Let the equation of the line be $y = kx + b$. Substituting the coordinates of points $A$ and $B$ into the equation, we obtain two equations:

$$-\sqrt{2} - 1 = k \cdot \frac{6 + \sqrt{2}}{2} + b$$

$$\sqrt{2} - 1 = k \cdot \frac{6 - \sqrt{2}}{2} + b$$

By subtracting the two equations to eliminate $b$, we get:

$$-2\sqrt{2} = k \cdot \sqrt{2}$$

Solving for $k$, we find $k = -2$. Substituting this value into the first equation, we get $b = 5$. Therefore, the equation of the line passing through point $M(3, -1)$ and bisecting the chord of the hyperbola is $\boxed{y = -2x + 5}$.

**Reference shortcut solution:** Let the points intersected with the chord be $A(x_1, y_1)$ and $B(x_2, y_2)$. Then, they satisfy the equation of the hyperbola:

$$\frac{x_1^2}{4} - y_1^2 = 1$$

$$\frac{x_2^2}{4} - y_2^2 = 1$$

Subtract the two equations:

$$\frac{x_1^2 - x_2^2}{4} - (y_1^2 - y_2^2) = 0$$

$$\frac{(x_1 - x_2)(x_1 + x_2)}{4} - (y_1 - y_2)(y_1 + y_2) = 0$$

Since $M(3, -1)$ is the midpoint of the chord, $x_1 + x_2 = 6, y_1 + y_2 = -2$. Substitute this into the above equation:

$$\frac{6(x_1 - x_2)}{4} + 2(y_1 - y_2) = 0$$

Then, we have the slope of the chord:

$$k = \frac{x_1 - x2}{y_1 - y_2} = \frac{3}{4}$$

Since $M(3, -1)$ is on the chord, $\boxed{3x + 4y - 5 = 0}$ is the line equation.

Table 12: Translated example of solutions from **GPT-4** and shortcut solutions. The red text is the reasoning step where the solution goes wrong.