# OpenReview forum: "Conic10K: A Challenging Math Problem Understanding and Reasoning Dataset"
_EMNLP/2023/Conference — EMNLP 2023 Findings_

### Official Review · Reviewer_4LaT · 2023-07-26

**Soundness:** 3

**Excitement:**

3: Ambivalent: It has merits (e.g., it reports state-of-the-art results, the idea is nice), but there are key weaknesses (e.g., it describes incremental work), and it can significantly benefit from another round of revision. However, I won't object to accepting it if my co-reviewers champion it.

**Paper Topic And Main Contributions:**

This paper proposes CONIC10K, a challenging math problem dataset on conic sections in Chinese senior high school education. The dataset contains various problems with different reasoning depths and only requires conic sections knowledge. Experiments on multiple models show that existing large language models, including GPT-4, exhibit weak performance on complex reasoning.

**Reasons To Accept:**

(1) This work proposes a 10k math problem dataset on conic sections including various problems with different reasoning depths.

(2) Detailed performance analysis of multiple state-of-the-art language models on subcategories of 10K datasets.


**Reasons To Reject:**

(1) Existing models have strong performance on semantic parsing, making this data less useful as a semantic parsing benchmark dataset.

(2) This work only explores zero-shot chain-of-thought without adding demonstrations. It will be good to know if adding more demonstrations can work.

(3) This paper lack of results of self-consistency [1], which sample multiple reasoning paths and conduct majority vote on the predictions.

Reference:
[1] Wang, Xuezhi, Jason Wei, Dale Schuurmans, Quoc Le, Ed Huai-hsin Chi and Denny Zhou. “Self-Consistency Improves Chain of Thought Reasoning in Language Models.” ArXiv abs/2203.11171 (2022): n. pag.

**Reproducibility:**

4: Could mostly reproduce the results, but there may be some variation because of sample variance or minor variations in their interpretation of the protocol or method.

**Reviewer Confidence:**

4: Quite sure. I tried to check the important points carefully. It's unlikely, though conceivable, that I missed something that should affect my ratings.

---

> ### Author Rebuttal · Authors · 2023-08-28
>
> Thank you for the review!
>
> > 1. Existing models have strong performance on semantic parsing, making this data less useful as a semantic parsing benchmark dataset.
>
> In Table 6, the mi-F1 and ma-F1 are indeed nearly perfect. However, to correctly understand a math problem, we must correctly understand every sentence in the problem -- it is almost impossible to solve a problem based on imperfect understanding. This indicates that the main metric of mathematical understanding should be accuracy instead of F1 scores. And in Table 6, we can find that the highest accuracy is 84%, which is far from being perfect yet (line 497). Our case study (line 446) also provides new insights on improving LLMs' ability to mathematical understanding.
>
> Furthermore, we would like to note that resources for math problem semantic parsing are very limited. Existing works containing formal language data are either not open-source [1], or too simple such that a simple string processing script is sufficient [2]. We believe that good performance of semantic parsing must rely on sufficient high-quality data, and that is what we create in this work.
>
>
> > 2. This work only explores zero-shot chain-of-thought without adding demonstrations. It will be good to know if adding more demonstrations can work.
> > 3. This paper lack of results of self-consistency, which sample multiple reasoning paths and conduct majority vote on the predictions.
>
> Thank you for the advice. After reading feedback from all reviewers, we decide to add experimental results of in-context learning and majority voting in our next version. These results are consistent with our previous observations, and they indeed better support our conclusions (line 493).
>
> The results are shown below:
>
> | Method                        | MathQA Acc |
> | ----------------------------- | ---------- |
> | GPT-3.5-turbo (Table 7)       | 6.2        |
> | GPT-3.5-turbo+in-context-learning        | 5.9        |
> | GPT-3.5-turbo+majority-voting | 6.8        |
>
> We find that for in-context learning, randomly adding samples from the train set slightly hurts the performance. By looking into the failure cases, we find the model is likely to copy the reasoning steps or numerical values from the few-shot demonstrations, which are often incorrect. Though it is possible that better in-context learning methods can improve performance marginally, we think the evidence is enough to support our conclusions.
>
>
>
> [1]: Sun, R., Zhao, Y., Zhang, Q., Ding, K., Wang, S., & Wei, C. (2019). A neural semantic parser for math problems incorporating multi-sentence information. ACM Transactions on Asian and Low-Resource Language Information Processing (TALLIP), 18(4), 1-16.
> [2]: Lu, P., Gong, R., Jiang, S., Qiu, L., Huang, S., Liang, X., & Zhu, S. C. (2021, August). Inter-GPS: Interpretable Geometry Problem Solving with Formal Language and Symbolic Reasoning. In Proceedings of the 59th Annual Meeting of the Association for Computational Linguistics and the 11th International Joint Conference on Natural Language Processing (Volume 1: Long Papers) (pp. 6774-6786).

---

### Official Review · Reviewer_As6G · 2023-08-04

**Soundness:** 4

**Excitement:**

3: Ambivalent: It has merits (e.g., it reports state-of-the-art results, the idea is nice), but there are key weaknesses (e.g., it describes incremental work), and it can significantly benefit from another round of revision. However, I won't object to accepting it if my co-reviewers champion it.

**Paper Topic And Main Contributions:**

This papers proposes CONIC10K, a challenging math problem dataset on conic sections in Chinese senior high school education, with high-quality annotations of formal representations.
Results show that existing LLMs have poor performance in mathematical reasoning, while most models could achieve good performance in mathematical understanding (semantic parsing).

**Reasons To Accept:**

1. This dataset labeled with formal representations is useful for LLMs reasoning research.
2. Paper is well-written and the idea is intuitive.

**Reasons To Reject:**

1. Lack of experiments on the relationship between semantic parsing and mathQA. For example, does the semantic parsing help mathQA?
2. Experimental analysis reveals little insight. For example, how can this dataset be used to help models improve their reasoning? What are possible future research directions?

**Reproducibility:**

4: Could mostly reproduce the results, but there may be some variation because of sample variance or minor variations in their interpretation of the protocol or method.

**Reviewer Confidence:**

4: Quite sure. I tried to check the important points carefully. It's unlikely, though conceivable, that I missed something that should affect my ratings.

---

> ### Author Rebuttal · Authors · 2023-08-28
>
> Thank you for the review!
>
> > 1. Lack of experiments on the relationship between semantic parsing and mathQA. For example, does the semantic parsing help mathQA?
>
> We would like to clarify that these two tasks, semantic parsing and mathQA, are not in a 'pipeline' relationship. Instead, they are two separate tasks that evaluate different abilities of LLMs (line 64). Our purpose is to figure out how well LLMs understand math problems and do mathematical reasoning respectively. Semantic parsing, in our work, is not designed to help mathQA.
>
> While it is beyond the scope of this paper, we have tried multi-task training to see if the performance could be improved. The ratio of training data from semantic parsing and mathQA was set to 3:2. Experimental results show that simple multi-task training hurts the performance of mathQA.
>
> | Method                       | Semantic Parsing Acc | MathQA Acc |
> | ---------------------------- | -------------------- | ---------- |
> | Bloomz-7b1+single-task | 71.3*                 | 15.4       |
> | Bloomz-7b1+multi-task        | 71.6                 | 12.3       |
>
> _*This result is different from that reported in Table 6. This is because we fixed a bug about tailing spaces due to the special tokenizer of Bloomz. We will update Bloomz-related results in Table 6 in our next version._
>
>
> > 2. Experimental analysis reveals little insight. For example, how can this dataset be used to help models improve their reasoning? What are possible future research directions?
>
> We would like to point out that our dataset is designed as a benchmark to evaluate models' ability of mathematical understanding and reasoning, just like the famous MATH dataset [1] (which serves as a benchmark for mathematical problem solving).
>
> Our results show that existing large language models have poor performance in mathematical reasoning, while most models could achieve good performance in mathematical understanding (but not perfect yet).
>
> We call for new LLMs (or any other methods, e.g., neural-symbolic methods) in the future to tackle the challenge of accurate mathematical understanding (math problems require absolute accurate understanding) and complex reasoning. As discussed in the introduction, this challenge has seldom been emphasized before because of a lack of dedicated data sources, and we present this challenge to the community.
>
> [1]: Hendrycks, D., Burns, C., Kadavath, S., Arora, A., Basart, S., Tang, E., Song, D., & Steinhardt, J. (2021). Measuring Mathematical Problem Solving With the MATH Dataset. NeurIPS.

---

### Official Review · Reviewer_WDny · 2023-08-07

**Soundness:** 4

**Excitement:**

4: Strong: This paper deepens the understanding of some phenomenon or lowers the barriers to an existing research direction.

**Missing References:**

See above
also
https://arxiv.org/abs/2209.09513
https://aclanthology.org/2020.acl-main.92/
https://aclanthology.org/N19-1246.pdf
https://aclanthology.org/2023.eacl-main.30/
https://aclanthology.org/K19-1033/

**Paper Topic And Main Contributions:**

The authors propose CONIC10K, a challenging math problem dataset on conic sections in Chinese senior high school education. This contains various problems with different reasoning depths, while only the knowledge from conic sections is required. For each problem, authors provide a formal representation, the reasoning steps, and the final solution. Experiments show that this benchmark is hard for GPT4.

**Questions For The Authors:**

See above

**Reasons To Accept:**

- interesting benchmark highlighting weakness of LLMs like GPT4
- inclusion of formal representation and reasoning steps has merit
- senior high school education content acts as default quality checker
- analysis in section 5 is interesting and useful

**Reasons To Reject:**

- I was looking for more insights on data quality analysis (beyond section 3.4), specially because the dataset is not available for us to look at some samples and get more idea about the data quality.

- how reliable is the evaluation? Is there a human study to gauge efficacy of the evaluation protocol of this benchmark?

- L443 "We inspect and analyse both success and failure cases in the experiment": did you select a subset of the benchmark? can you describe more how you did this evaluation?

- L276: instruction tuning: did you try detailed instructions than the ones in Table 5? Since this is a hard benchmark requiring specialized knowledge, detailed instructions in the style of https://aclanthology.org/2022.acl-long.244.pdf might be helpful to improve models.

**Reproducibility:**

2: Would be hard pressed to reproduce the results. The contribution depends on data that are simply not available outside the author's institution or consortium; not enough details are provided.

**Reviewer Confidence:**

4: Quite sure. I tried to check the important points carefully. It's unlikely, though conceivable, that I missed something that should affect my ratings.

---

> ### Author Rebuttal · Authors · 2023-08-28
>
> Thank you for the review! Glad to have your helpful feedback!
>
>
> > 1. I was looking for more insights on data quality analysis (beyond section 3.4), specially because the dataset is not available for us to look at some samples and get more idea about the data quality.
>
> Sorry, but we are confused about what you mean by 'data quality':
> 1. If it means the annotation quality, then this is described in section 3.3.2 and Appendix A.3. We apply multiple methods to ensure the annotation quality (this is what we are proud of).
> 2. If it means the data source quality, then this is described in section 3.3.1. All the questions come from real-world exams or exercises. We also ask annotators to manually check all the question texts and rule out incomplete or repeated questions.
> 3. If it means the diversity of the dataset, then it is related to the data source. Since we collect data from real-world scenarios instead of using templates/model generation, our data distribution is consistent with real-world applications. We looked into the references you provided and found that some analysis methods like n-grams may not suit our work. This is because our dataset is not an MWP dataset. Instead, it uses a closed vocabulary and all questions are expressed in the mathematical language. So if you do n-gram, you will find that many natural language phrases are quite similar (we did this, only to find over 75% of the questions start with '则'). The actual difference, or diversity, lies in the information provided by the question (or the reasoning path required).
> After looking into the references you provided, we decide to add the following statistics in our next version: 1) a word cloud that has been translated to English; 2) a question length distribution. We count each latex command as one single character. The figures are shown in the links below:
> https://i.imgur.com/Gy38zI5.png
> https://i.imgur.com/JlfjuuI.png
> 4. It is possible that our guesses above are all wrong. If so, please point it out and we are happy to discuss this further.
>
> We provide a few samples in Figure 1 and Table 9-11. We will add more samples in the Appendix in the next version. Meanwhile, the readers can always visit our GitHub link and check the entire dataset after the publication of this paper.
>
> > 2. how reliable is the evaluation? Is there a human study to gauge efficacy of the evaluation protocol of this benchmark?
>
> We perform 2 tasks in our experiments: semantic parsing and mathQA.
> - For mathQA, we simply do human evaluation (section 4.4.2). We manually compare the answers and reference solutions. Since the reference solution is just a number, math expression or natural language phrase, the evaluation is totally objective and accurate.
> - For semantic parsing, we develop a script to automatically evaluate the results (section 4.4.1). When the project is initiated (line 194), we find that it is very difficult for humans to recognize the mistakes in formal representation, especially when the prediction is very close to the reference. Script evaluation is much faster and more accurate than other methods, including human evaluation. During the annotation stage, we use the same script to help the validator compare two annotations (line 213), and the feedback shows that the evaluation is accurate and reliable. This is the best evaluation method we could come up with except for spending lots of time to build a logic reasoner and compare the final result (which might also introduce false positive results, where the formal representation is incorrect while the final result accidentally hits the reference solution).
>
> > 3. L443 "We inspect and analyse both success and failure cases in the experiment": did you select a subset of the benchmark? can you describe more how you did this evaluation?
>
> This is just a case study. We follow these steps:
> 1. We inspect all failure cases (both the semantic parsing and the mathQA tasks), manually checking them one by one;
> 2. We try to conclude some common features from these failure cases;
> 3. We confirm these features by inspecting the success cases, ensuring that these features do not appear in the success cases.
>
>
> > 4. L276: instruction tuning: did you try detailed instructions than the ones in Table 5? Since this is a hard benchmark requiring specialized knowledge, detailed instructions in the style of https://aclanthology.org/2022.acl-long.244.pdf might be helpful to improve models.
>
> Thank you for the advice. After reading feedback from all reviewers, we decide to add the results of instruction tuning with detailed instructions in our next version. The detailed instruction consists of task name, task definition, and examples. The results are consistent with our previous observations, and they indeed better support our conclusions (line 493).
>
> The results are shown below:
>
> | Method                                       | MathQA Overall Accuracy |
> | -------------------------------------------- | ----------------------- |
> | Bloomz-7b1+LoRA+simple instruction (Table 7) | 15.4                    |
> | Bloomz-7b1+LoRA+detailed instruction         | 15.7                    |

---

### Meta-Review · Area_Chair_UJsx · 2023-09-16

**Recommendation:** 4

**Metareview:**

**Summary:**
The authors introduce CONIC10K, a challenging dataset consisting of mathematics problems focused on conic sections in Chinese senior high school education. The dataset encompasses a set of problems having different levels of reasoning complexity, all requiring a basic knowledge limited to conic sections. For each problem, the authors give a formal representation, outline the reasoning steps, and present the final solution. Experimental results across multiple models demonstrate that even state-of-the-art large language models like GPT-4 exhibit mediocre performance when confronted with intricate reasoning tasks.

**Strengths:**
The reviewers are in consensus on the paper's strengths, which can be summarized as follows:
1. The paper introduces a 10,000-problem dataset on conic sections, encompassing a wide range of problems with different levels of reasoning complexity. This dataset serves as an interesting benchmark, shedding light on the limitations of Large Language Models (LLMs) such as GPT-4.
2. The paper includes a comprehensive performance analysis of several state-of-the-art language models across subcategories of the 10,000-problem dataset.
3. An additional noteworthy aspect is that the senior high school education content serves as a quality checker.

**Weaknesses:**
Reviewers are in consensus that the paper lacks insights into data quality analysis, especially given the unavailability of the dataset for sample examination. Questions also arise regarding the reliability of the benchmark's evaluation protocol. Additionally, the study lacks experiments investigating the relationship between semantic parsing and mathQA, as existing models excel in semantic parsing, diminishing the dataset's utility as a semantic parsing benchmark. Furthermore, it would be beneficial to explore the effectiveness of adding more demonstrations. Lastly, the paper lacks results regarding self-consistency, which involve sampling multiple reasoning paths and conducting a majority vote on predictions.

**Author-Reviewer discussion and acknowledgment:**
Reviewers presented comments and questions, all of which the authors addressed in their rebuttal response. All reviewers acknowledged the authors' rebuttal.

**Conclusion:**
The paper is well-written and the idea is intuitive. However, the reviewers recommend that the authors make further improvements to enhance the paper and incorporate missing references into the bibliography.

---

### Decision · Program_Chairs · 2023-10-07

**Decision:**

Accept-Findings

**Comment:**

**Summary:**
The authors introduce CONIC10K, a challenging dataset consisting of mathematics problems focused on conic sections in Chinese senior high school education. The dataset encompasses a set of problems having different levels of reasoning complexity, all requiring a basic knowledge limited to conic sections. For each problem, the authors give a formal representation, outline the reasoning steps, and present the final solution. Experimental results across multiple models demonstrate that even state-of-the-art large language models like GPT-4 exhibit mediocre performance when confronted with intricate reasoning tasks.

**Strengths:**
The reviewers are in consensus on the paper's strengths, which can be summarized as follows:
1. The paper introduces a 10,000-problem dataset on conic sections, encompassing a wide range of problems with different levels of reasoning complexity. This dataset serves as an interesting benchmark, shedding light on the limitations of Large Language Models (LLMs) such as GPT-4.
2. The paper includes a comprehensive performance analysis of several state-of-the-art language models across subcategories of the 10,000-problem dataset.
3. An additional noteworthy aspect is that the senior high school education content serves as a quality checker.

**Weaknesses:**
Reviewers are in consensus that the paper lacks insights into data quality analysis, especially given the unavailability of the dataset for sample examination. Questions also arise regarding the reliability of the benchmark's evaluation protocol. Additionally, the study lacks experiments investigating the relationship between semantic parsing and mathQA, as existing models excel in semantic parsing, diminishing the dataset's utility as a semantic parsing benchmark. Furthermore, it would be beneficial to explore the effectiveness of adding more demonstrations. Lastly, the paper lacks results regarding self-consistency, which involve sampling multiple reasoning paths and conducting a majority vote on predictions.

**Author-Reviewer discussion and acknowledgment:**
Reviewers presented comments and questions, all of which the authors addressed in their rebuttal response. All reviewers acknowledged the authors' rebuttal.

**Conclusion:**
The paper is well-written and the idea is intuitive. However, the reviewers recommend that the authors make further improvements to enhance the paper and incorporate missing references into the bibliography.